# Changes in SGLT2i and GLP-1RA real-world initiator profiles following cardiovascular outcome trials: A Danish nationwide population-based study

Jakob S. Knudsen[1]*, Lisbeth M. Baggesen[1], Maria Lajer[2], Larisa Nurkanovic[3], Anastasia Ustyugova[4], Henrik T. Sørensen[1], Reimar W. Thomsen[1]

1 Department of Clinical Epidemiology, Aarhus University Hospital, Aarhus, Denmark, 2 Boehringer Ingelheim, Copenhagen, Denmark, 3 Boehringer Ingelheim, Amsterdam, Netherlands, 4 Boehringer Ingelheim International GmbH, Ingelheim am Rhein, Germany

* jsk@clin.au.dk

## Abstract

### Background

We investigated changes in clinical characteristics of SGLT2i and GLP-1RA real-world initiators in Denmark before/after landmark cardiovascular outcome trials.

### Methods

We compared first-time SGLT2i (25,070) and GLP-1RA (14,671) initiators to initiators of DPP-4i (n = 34,079), a class without proven cardiovascular benefits. We used linked population-based healthcare data to examine initiation incidence, medication patterns, and pre-existing atherosclerotic cardiovascular disease (ASCVD) during 2014–2017.

### Results

Nationwide incidence of SGLT2i initiators increased 3.6-fold (53/100,000 to 172/100,000 per year) vs. a 1.5-fold increase for GLP-1RA. DPP-4i initiation remained stable. From the end of 2015, SGLT2i was increasingly used as 2nd-line therapy, while medication patterns were much more stable for GLP-1RA. Among SGLT2i users, ASCVD increased slightly from 28% to 30%; age- and gender-adj. prevalence ratio (aPR) = 1.03 (95% CI:0.97–1.10). In contrast, among GLP-1RA initiators, baseline ASCVD declined from 29% to 27% (aPR: 0.90 (95% CI:0.84–0.97)), and in DPP-4i initiators from 31% to 29% (aPR: 0.91 (95% CI:0.88–0.96)).

### Conclusions

Following the EMPA-REG OUTCOME trial in 2015, SGLT2i have become increasingly used as 2nd-line treatment in everyday clinical practice, with only minor increases in patient proportions with ASCVD. For GLP-1RA, we observed more stable therapy lines and slightly decreasing ASCVD in new users despite the LEADER trial.

**Data Availability Statement:** Due to restrictions related to Danish law and protecting patient privacy, the combined set of data as used in this

study can only be made available through a trusted third party, Statistics Denmark. This state organisation holds the data used for this study. University-based Danish scientific organisations can be authorized to work with data within Statistics Denmark and such organisation can provide access to individual scientists inside and outside of Denmark. Requests for data may be sent to Statistics Denmark: forskningsservice@dst.dk or +45 39 17 31 32.

**Funding:** Sources of funding: This work was supported by an institutional research grant from Boehringer Ingelheim to Aarhus University. While Boehringer Ingelheim was involved in the study's concept, it was designed and conducted by the coauthors from Aarhus University. The funder provided support in the form of salaries for authors ML, LN, and AU, but did not have any additional role in the study design, data collection and analysis, decision to publish, or preparation of the manuscript. The specific roles of these authors are articulated in the 'author contributions' section.

**Competing interests:** Conflict of interest disclosures: ML, LN and AU are employees of Boehringer Ingelheim. The other authors have no personal conflicts of interest relevant to this article. The Department of Clinical Epidemiology is involved in studies with funding from various companies as research grants to (and administered by) Aarhus University. The commercial affiliation did not alter our adherence to PLOS ONE policies on sharing data and materials. However, Danish law does not allow researchers to share raw data from the registries with third parties. Data can be accessed by researchers through application to the Danish Data Protection Agency and the Danish Health Data Authority

**Abbreviations:** ASCVD, atherosclerotic cardiovascular disease; ATC, Anatomical Therapeutic Chemical; CCI, Charlson comorbidity index; CI, Confidence Interval; CPR, Central Personal Registry; CRS, Civil Registration System; CV, cardiovascular; DNPR, Danish National Patient Register; DPP-4i, dipeptidyl peptidase 4 inhibitors; GLP-1RAs, glucagon-like peptide-1 receptor agonists; HF, Heart failure; ICD-10, International Statistical Classification of Diseases and Related Health Problems 10th revision; MI, Myorcardial Infarction; Q, Quarter; SGLT2i, Sodium-glucose co-transporter 2 inhibitors; T2D, type 2 diabetes.

## Introduction

Sodium-glucose co-transporter 2 inhibitors (SGLT2i) represent a new and increasingly used class of oral antihyperglycemic drugs for type 2 diabetes (T2D) [1]. These agents currently include 4 agents: dapagliflozin, canagliflozin, empagliflozin, ertugliflozine, and combinations of these SGLT2i with metformin or other antihyperglycemic drugs [2–4]. Similarly, glucagon-like peptide-1 receptor agonists (GLP-1RAs), currently including exenatide, liraglutide, dulaglutide, lixisenatide, semaglutide, and paired injectables in combinations with other antihyperglycemic drugs, as a class are increasingly used in the treatment of T2D over the last decade [5–7]. While metformin has remained the recommended initial antihyperglycemic drug for most patients with T2D, international (and Danish) guidelines until now have recommended a free choice among several second or third line treatment options, based on an individualised treatment approach [8]. In the most recent years, the prescription patterns of SGLT2i and GLP-1RAs in real-world settings may have been increasingly influenced by landmark cardiovascular (CV) outcome trials instigated by regulatory authorities to promote patient safety [9], but data on incident utilization trends are scarce. In 2015, the empagliflozin EMPA-REG OUTCOME trial [10] showed a reduced risk of CV outcomes, CV mortality, and all-cause mortality in patients with T2D with existing CV disease. In 2017, the canagliflozin CANVAS trial program [11] showed a reduced risk of major adverse CV events in patients with T2D and high CV risk. For GLP-1RAs, the 2016 liraglutide LEADER trial showed a reduced risk of CV outcomes, CV mortality, and all-cause mortality in patients with high CV risk. A reduced risk of CV outcomes was also observed in similar patients receiving semaglutide in the SUSTAIN-6 trial in 2016 [12,13]. Therefore, in the most recent updates to the EASD/ADA and Danish guidelines from 2018 and 2019 [14–16], initiation of a SGLT2i or a GLP-1RA with proven CV benefit has been recommended for patients with T2D and clinical CV disease, with currently strongest evidence available for liraglutide and empagliflozin [14–16].

There are scarce population-based data on how the initiation rates and clinical profiles of initiators of SGLT2i or GLP-1RA have evolved in real-world settings before and after publication of key trial results [3–7,9]. Linked Danish population-based healthcare databases provide a unique opportunity to characterize recent SGLT2i and GLP-1RAs utilization trends in Denmark, and to clinically describe all individuals with incident use of these drugs.

We therefore aimed to examine trends in initiation incidence rates, medication patterns at baseline, and baseline patient characteristics at the time of first drug initiation among SGLT2i and GLP-1RA new users, focusing on changes from 2014 to 2017. We compared the results to time trends for new DPP-4i users, a drug class without proven CV benefit. We hypothesized that publication of key CV outcome trial results and new drug labels during 2014–2017 may have influenced both the overall new SGLT2i and GLP-1RA user incidence, increased the proportion of patients who initiate these agents early in the course of diabetes (e.g., as second-line drugs after metformin), and increased the proportion who had preexisting atherosclerotic cardiovascular disease (ASCVD) at the time of drug initiation.

## Materials and methods

### Setting and source population

We conducted nationwide cross-sectional analyses of linked Danish population-based healthcare databases to characterize all initiators of SGLT2i and GLP-1RA in Denmark during 2014 through 2017. We first identified a source population consisting of all individuals who lived in Denmark and redeemed a prescription of an antihyperglycemic drug in the period 1995–2017, defined as filling one or more prescriptions for: metformin, sulfonylurea, thiazolidinedione,

SGLT2i, GLP-1RA, DPP-4i, insulin, alpha-glucosidase inhibitor, other oral antihyperglycemic drugs, or combination products, according to the Anatomical Therapeutic Chemical (ATC) classification system (codes A10A, A10B). [17]. Diabetic patients who under the age of 30 used insulin as mono-therapy and never used oral antihyperglycemic medications were excluded as likely type 1 diabetes patients [18,19]. The remaining individuals were defined as having T2D. Within this population of incident initiators of antihyperglycemic drugs for T2D 1995–2017, all incident first-time users of SGLT2i, GLP-1RA, and DPP-4i in the period 1 January 2014 to 31 December 2017 (i.e., no previous use of the respective drug recorded between 1995 and 2014) were identified. In our main analysis, we disregarded initiation of the GLP-1RA liraglutide brand-named Saxenda® as an inclusion criterion for the GLP-1RA initiator cohort, because Saxenda® (liraglutide 3.0 mg daily) was approved as a treatment for chronic weight management in obese patients in 2015. In an additional sensitivity analysis, we also included Saxenda® initiators in the GLP-1RA initiator cohort.

## Data sources

The Civil Registration System (CRS) holds records of central personal registry (CPR)-number, address, marital status, emigration and immigration status, and date of death (if any) of the entire population of Denmark (current population 5.7 mio) since 1968. This system can be used to unambiguously link all Danish registries containing CPR-numbers [20]. The Danish National Patient Register (DNPR) includes information of all hospitalized patients since 1977 and on outpatient hospital contacts since 1995. The register contains information about the date of admission, discharge, diagnosis codes and surgical procedures. From 1994 onwards they have been coded according to International Statistical Classification of Diseases and Related Health Problems 10th revision (ICD-10) [21]. The Danish National Prescription Registry covers all drug prescriptions redeemed at any pharmacy in Denmark since 1995, including patient's identifier, date of sale, type of drug, and universal product number (*Varenummer*), which encodes medication name [22]. Computerized clinical biochemistry test results have been kept in the LABKA Database for all samples taken in primary or secondary care among persons living in North and Central Denmark (apprx 30% of the total Danish population) beginning in 1997 and complete from 2000 [17,23,24].

## Characteristics of SGLT2i, GLP-1RA, and DPP-4i initiators

For all patients with a first prescription of SGLT2i, GLP-1RA, or DPP-4i in 2014–2017 (the index date), either as their first ever antihyperglycemic drug prescription or as intensification or replacement therapy for previous antihyperglycemic drug use (e.g. metformin), we ascertained data on age, gender, place of residence, and marital status. Using the DNPR, we assessed a complete hospital contact history for each individual for any previous hospital-diagnosed ASCVD, both overall and for individual conditions (i.e., atherosclerotic heart disease including myocardial infarction, angina pectoris, or any coronary surgery; atherosclerotic cerebrovascular disease including stroke, TCI, or thrombolysis/thrombectomy; or peripheral vascular disease including claudication, vascular surgery, extremity amputation procedures). We also assessed pre-existing hospital-diagnosed heart failure, renal disease, medical obesity, and a number of other important comorbidities including COPD, cancer, liver disease, alcoholism-related conditions, and previous infections (see S1 File for codes). We assessed the comorbidity burden using the Charlson comorbidity index (CCI) [25], and calculated a total score for each patient (no comorbidities [score = 0], moderate comorbidity burden [score = 1], severe comorbidity burden [score = 2] or very severe comorbidity burden [score > 2]). We further assessed use of any comedications on the index date (type and number of other antihyperglycemic

therapies, any CV medications including antihypertensives, antiplatelet therapy, or lipid-lowering drugs, and glucocorticoids), and diabetes duration on the index date (years since first ever recorded diabetes hospital diagnosis or diabetes therapy start). For the regional subcohort in North and Central Denmark with available laboratory data (~30% of the T2D population), we also ascertained $HbA_{1c}$ (last measured value within 12 months), eGFR based on last measured creatinine (calculated using CKI-EPI equation [26]), and LDL cholesterol values.

## Ethics

The study was approved by the Danish Data Protection Agency. Analyses were conducted on pseudonymized data at the Danish Health Data Authority. The study was purely registry-based and did not involve any contact with patients or interventions; therefore, according to Danish legislation, no informed consent or approval from the health research ethics committee was required.

## Statistical methods

For all graphical time trend analyses, we plotted dates (quarter) of the following events on our timeline: TECOS: Sitagliption (DPP-4i) showed non-inferiority to placebo (June 2015) [25]; Obesity label: Liraglutide 3.0 mg daily launched as treatment for obesity (August 2015); EMPA-REG OUTCOME: empagliflozin showed CV and CV/all-cause mortality benefits (September 2015) [4], LEADER: liraglutide showed CV and CV/all-cause mortality benefits (June 2016) [6]; CANVAS: canagliflozin showed CV benefits (June 2017) [5]; Empagliflozin launched as treatment in T2D patients with CVD (CV label) (January 2017); Liraglutide launched as treatment in T2D patients with CVD (CV label) (June 2017).

## Initiation incidence

Firstly, based on redeemed prescriptions 1995–2017, we calculated and graphically plotted the number of first ever users of SGLT2i, GLP-1RA, or DPP-4i with 95% confidence intervals, per 100,000 inhabitants in Denmark for each calendar year, of each of the study medications by calendar year and quarter from 2014–2017. We examined increases between calendar year 2014 and 2017 in incident users of study medications per 100,000 person-years. We repeated these analysis and plots for individual drugs, within the drug classes.

## Baseline medication patterns

Secondly, based on redeemed antihyperglycemic therapy prescriptions 100 days prior to the index date (typical pack size of antihyperglycemic drugs in Denmark 90 to 100 daily doses), we calculated and graphically plotted the proportions who, prior to initiating treatment with SGLT2i, GLP-1RA, or DPP-4i, received: no antihyperglycemic therapy, monotherapy, dual therapy, and triple or higher (multiple) therapy, by calendar year and quarter from 2014–2017.

## Baseline characteristics

Thirdly, for each of the index drugs, we calculated the proportion of all initiators in the total study period having each of the baseline characteristics. We calculated prevalence ratios comparing GLP-1RA and SGLT2i with DPP-4i initiators as a common reference group, using modified Poisson regression to adjust for age and sex (in order to be able to evaluate if a difference in prevalence ratios was more than could be attributed to the difference arising from age and sex difference found between the groups). Next, we examined if proportions with ASCVD and other important patient characteristics at initiation changed in timely relation to CV

outcome trial publication, new drug labels, or other main events during the period 2014–2017. Changes in age- and gender-adjusted prevalence ratios (aPRs) of characteristics within each drug class internally, using the last study year 2017 versus 2014 as the reference year was calculated. For selected pre-defined characteristics of special interest (any ASCVD, HF, stroke, MI, and hospital-coded medical obesity) we graphically plotted the evolving proportion of initiators with these characteristics for each of the three study medication classes, with 95% confidence intervals, by calendar year and quarter from 2014–2017. For the same time periods, we calculated the proportion (with 95% confidence intervals) of patients that had redeemed a prescription for antihypertensive treatment and statins one year prior to initiation of an index drug. Since Saxenda® (liraglutide 3.0 mg daily, obesity treatment label) may be sometimes used in clinical practice as antihyperglycemic treatment in patients who are both obese and have T2D, we did a sensitivity analysis also including patients iniating Saxenda® in the GLP1-RA cohort.

## Results

We identified 25,070 new first-time SGLT2i initiators, 14,671 first-time GLP-1RA initiators, and 34,079 first-time DPP-4i initiators in Denmark during 2014–2017. As a point of comparison, the total number of unique prevalent users in our data during 2014–2017 was 32,091 for SGLT2i, 37,282 for GLP-1RA, and 64,613 for DPP-4i. During the four years study period, the incidence of SGLT2i initiators increased 3.6-fold, from 53 / 100,000 person-years (PY) in 2014 to 172 / 100,000 PY in 2017 (quarterly changes can be seen in Fig 1). In comparison, the number of GLP-1RA initiators increased 1.5-fold, while the number of DPP-4i initiators remained very stable (1.05-fold increase) throughout 2014–2017 (Fig 1). Liraglutide was almost exclusively used in the GLP-1RA class throughout 2014–2017, while empagliflozin quickly became the most commonly prescribed SGLT2i following the EMPAREG outcome trial (S1 File).

Fig 2 shows that early on during the 2014–2017 period, SGLT2i was most often prescribed as third-line treatment; however, the likelihood of initiating SGLT2i as second-line therapy increased substantially between 2014 and 2017. Use of SGLT2i in patients previously receiving antihyperglycemic monotherapy increased from 22% in 2015 Q3 to 36% in 2017 Q4. In comparison, initiation lines were rather stable for GLP-1RA initiators during 2014–2017 (Fig 2). When also including patients iniating Saxenda® (liraglutide 3.0 mg daily, obesity treatment label) in the analysis, GLP-1RA use as monotherapy increased from 2015 and onwards (S1 File). DPP-4i were predominantly and consistently used as second-line therapy throughout the study period (Fig 2).

Overall, prevalence of common diabetes complications and other comorbidities at drug initiation was rather similar between the three drug groups. For example, a history of peripheral vascular disease was present in 8.7% of SGLT2i, 8.5% of GLP-1RA, and 8.3% of DPP-4i initiators. However, patients who initiated SGLT2i (median age 61 years [IQR 53–69]) or GLP-1RA (59 years [IQR 51–68]) were younger than DPP-4i initiators (66 years [IQR 56–74]) (Table 1). At the same time, SGLT2i initiators had a longer diabetes history at baseline (median 8.0 years [IQR 4.6–12.4]) than GLP-1RA (6.7 years [IQR 3.3–11.2] or DPP-4i initiators (5.4 years [IQR 2.1–9.6]) (Table 1). During 2014–2017, any atherosclerotic ASCVD at baseline was present in 29% of SGLT2i initiators, 28% of GLP-1RA initiators, and 30% of DPP-4i initiators. After controlling for differences in gender and in particular for the younger age in SGLT2i and GLP-1RA initiators, this corresponded to adjusted prevalence ratios (aPRs) for ASCVD of 1.09 (95% CI 1.06–1.12) for SGLT2i and 1.13 (95% CI 1.10–1.16) for GLP-1RA initiators, versus the reference group of DPP-4i initiators. Prevalence proportions and aPRs for ischemic heart

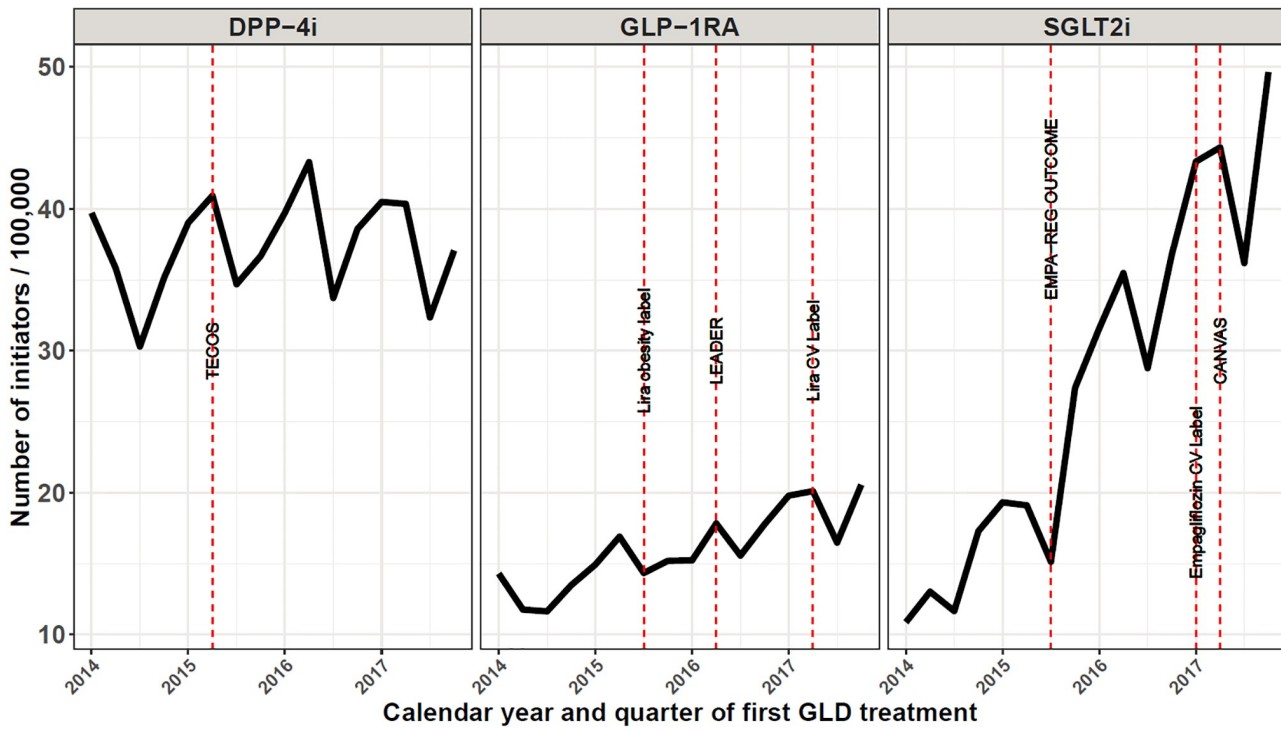

**Fig 1. Quarterly number of initiators of DPP-4i, GLP-1RA, and SGLT2i in Denmark, 2014–2017.** DPP-4i: dipeptidyl peptidase-4 inhibitor; GLP-1RA: glucagon-like peptide-1 receptor agonists; SGLT2i: sodium-glucose cotransporter 2 inhibitors; TECOS: Sitagliptin (DPP-4i) showed non-inferiority to placebo [35]; Lira obesity label: Liraglutide 3 mg launched as treatment for obesity; EMPA-REG OUTCOME: empagliflozin showed CV and CV/all-cause mortality benefits [10], LEADER: liraglutide showed CV and CV/all-cause mortality benefits [12]; CANVAS: canagliflozin showed CV benefits [11].

disease, cerebrovascular disease, heart failure, and for a number of other comorbidities at baseline are also shown in Table 1.

Fig 3 shows time trends from 2014 Q1 to 2017 Q4 in the proportion of SGLT2i, GLP-1RA, and DPP-4i initiators who had established ASCVD and hospital-diagnosed obesity, respectively, at treatment initiation. For SGLT2i initiators, the proportion with any ASCVD increased slightly from 28% in 2014 to 30% in 2017, partly related to increasing patient age over time, which corresponded to an aPR of 1.03 (95% CI: 0.97–1.10) in 2017, as compared with the first study year 2014 (S1 File). A slight increase in ASCVD since 2015 was seen in parallel in the two major groups of empagliflozin and dapagliflozin initiators, with the ASCVD proportion continuously being about 5 percentage points higher in empagliflozin than in dapagliflozin starters (S1 File). For GLP-1RA initiators, the ASCVD proportion was 29% in 2014 versus 27% in 2017 (aPR in 2017: 0.90 (95% CI: 0.84–0.97)) (S1 File). For DPP-4i initiators ASCVD also decreased slightly, from 31% to 29% (aPR in 2017: 0.91 (95% CI: 0.88–0.96) (S1 File). For SGLT2i initiators, hospital-diagnosed obesity declined from 29% to 24% while prevalence of obesity among GLP-1RA initiators increased from 27% to 32% (Fig 3). Prevalence of obesity among DPP4i initiators increased over the study period from 16% (2014) to 19% (2017), but remained clearly lower, compared to SGLT2s and GLP1 initiators.

Fig 4 shows time trends from 2014 Q1 to 2017 Q4 in the proportion of SGLT2i, GLP-1RA, and DPP-4i initiators who had acute myocardial infarction (AMI) and heart failure (HF), respectively, diagnosed prior to treatment initiation. For SGLT2i initiators, the proportion with any ischemic heart disease increased from 19% in 2014 to 22% in 2017, corresponding to

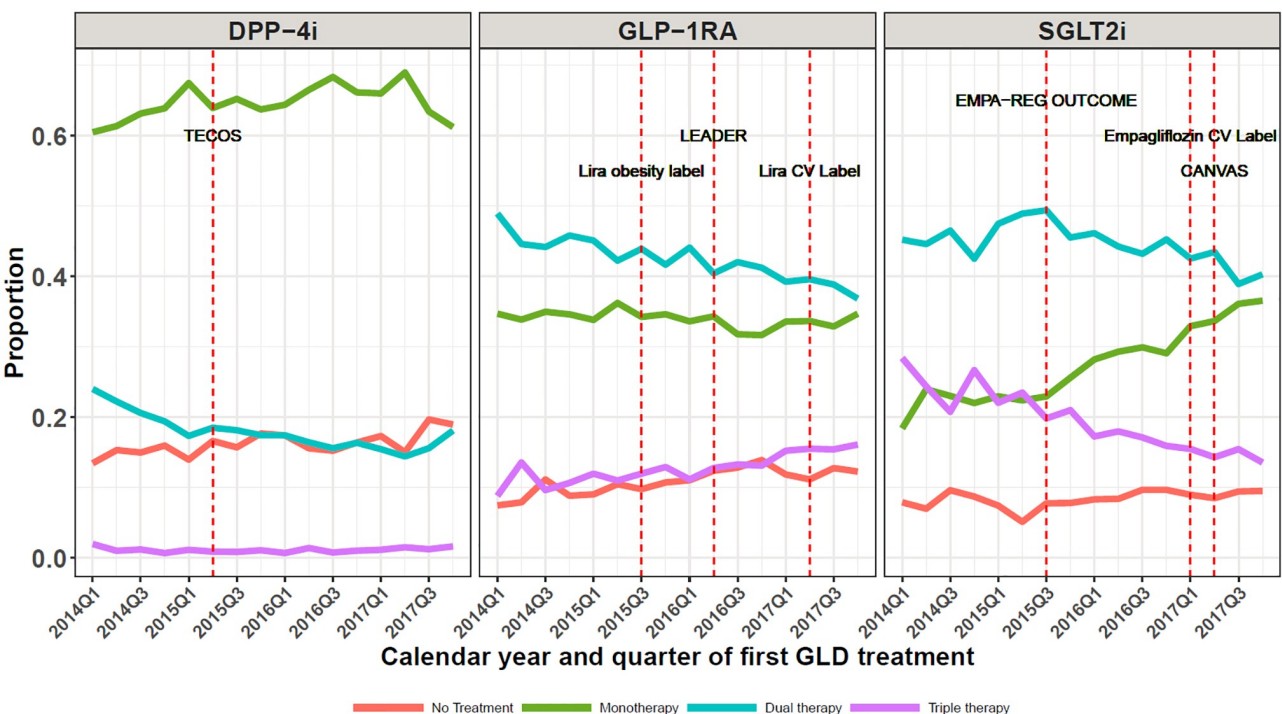

**Fig 2. Time trends in proportions using various baseline glucose-lowering drug regimens at the time of initiation of DPP-4i, GLP-1RA, and SGLT2i, respectively.** Graphs show the proportion of patients who–at the time of their first initiation of either DPP-4i, GLP-1RA, or SGLT2i –are on no other GLD treatment (red graph), on another GLD monotherapy (green graph), on dual GLD therapy (blue graph), or on triple GLD therapy (purple graph). DPP-4i: dipeptidyl peptidase-4 inhibitor; GLP-1RA: glucagon-like peptide-1 receptor agonists; SGLT2i: sodium-glucose cotransporter 2 inhibitors; TECOS: Sitagliptin (DPP-4i) showed non-inferiority to placebo [35]; Lira obesity label: Liraglutide 3 mg launched as treatment for obesity; EMPA-REG OUTCOME: empagliflozin showed CV and CV/all-cause mortality benefits [10], LEADER: liraglutide showed CV and CV/all-cause mortality benefits [12]; CANVAS: canagliflozin showed CV benefits [11].

an aPR of 1.10 (95% CI 1.01–1.19) in 2017 versus 2014 (S1 File). In contrast for GLP-1RA initiators and DPP-4i initiators, ischemic heart disease decreased slightly over time (S1 File). Similarly for established heart failure, the prevalence among SGLT2i initiators increased over time, from 5% to 7% (aPR 1.22 (95% CI 1.03–1.45)), while heart failure among GLP1-1RA and DPP-4i initiators decreased slightly. S1 File shows time trends from 2014 Q1 to 2017 Q4 in the proportion of SGLT2i, GLP-1RA, and DPP-4i initiators who had redeemed a prescription for antihypertensive drugs and statins 12 months prior to initiation. While the proportion with antihypertensive drug use was rather stable, the proportion using statins showed a declining tendency, especially in GLP-1RA initators.

## Discussion

In this real-life clinical care study, we observed many similarities in baseline patient characteristics between SGLT2i and GLP-1RA initiators, while differentially evolving time trends were observed for therapy lines of SGLT2i and GLP-1RA. Moreover, among SGLT2i initiators the proportions with established ASCVD or heart failure increased slightly between 2014 and 2017, while these proportions remained stable or decreased slightly among initiators of GLP-1RA or DPP-4i.

### Interpretation

**Incidence.** The much higher rise in incidence of SGLT2i versus GLP-1RA initiation during 2014 to 2017 may reflect that SGLT2i was a new drug class on the market in 2014, as

**Table 1. Clinical characteristics among real-world initiators of GLP-1RA, SGLT2i and DPP-4i in Denmark, 2014–2017.**

| | GLP-1RA | | | SGLT2i | | | DPP-4i | | |
|---|---|---|---|---|---|---|---|---|---|
| | N = 14,671 | Percent (%) | aPR (95% CI) versus DPP-4i § | N = 25,070 | Percent (%) | aPR (95% CI) versus DPP-4i § | N = 34,079 | Percent (%) | |
| **Sex** | | | | | | | | | aPR (95% CI) |
| female | 6435 | 44.2 | 1.11 (1.08–1.13) | 9621 | 38.4 | 0.96 (0.94–0.98) | 13628 | 40.0 | 1.00 (ref) |
| male | 8136 | 55.8 | 0.93 (0.91–0.94) | 15449 | 61.6 | 1.03 (1.02–1.04) | 20451 | 60.0 | 1.00 (ref) |
| **Age** | | | | | | | | | |
| 0–29 | 270 | 1.9 | 3.12 (2.60–3.74) | 162 | 0.6 | 1.12 (0.91–1.38) | 198 | 0.6 | 1.00 (ref) |
| 30–59 | 7157 | 49.1 | 1.42 (1.39–1.45) | 11303 | 45.1 | 1.30 (1.28–1.33) | 11788 | 34.6 | 1.00 (ref) |
| 60–69 | 4351 | 29.9 | 1.06 (1.03–1.10) | 8042 | 32.1 | 1.13 (1.11–1.16) | 9611 | 28.2 | 1.00 (ref) |
| 70+ | 2793 | 19.2 | 0.52 (0.50–0.54) | 5563 | 22.2 | 0.61 (0.59–0.62) | 12482 | 36.6 | 1.00 (ref) |
| Median Age (IQR) | 59 | (51–68) | | 61 | (53–69) | | 66 | (56–74) | 1.00 (ref) |
| **Region of residence** | | | | | | | | | |
| Capital Region | 4160 | 28.5 | 0.96 (0.93–0.99) | 7706 | 30.7 | 1.03 (1.00–1.05) | 10215 | 30.0 | 1.00 (ref) |
| Central Denmark Region | 3218 | 22.1 | 0.99 (0.95–1.03) | 5429 | 21.7 | 0.97 (0.94–1.00) | 7549 | 22.2 | 1.00 (ref) |
| North Denmark Region | 1433 | 9.8 | 0.87 (0.82–0.92) | 2428 | 9.7 | 0.85 (0.81–0.90) | 3889 | 11.4 | 1.00 (ref) |
| Region Zealand | 2512 | 17.2 | 1.14 (1.09–1.19) | 4355 | 17.4 | 1.14 (1.10–1.18) | 5211 | 15.3 | 1.00 (ref) |
| Southern Denmark | 3248 | 22.3 | 1.04 (1.00–1.08) | 5152 | 20.6 | 0.97 (0.94–1.00) | 7215 | 21.2 | 1.00 (ref) |
| **Diabetes duration** | | | | | | | | | |
| 0 days | 442 | 3.0 | 0.68 (0.61–0.75) | 248 | 1.0 | 0.22 (0.19–0.25) | 1375 | 4.0 | 1.00 (ref) |
| 0-<2 year | 2025 | 13.9 | 0.56 (0.54–0.59) | 2540 | 10.1 | 0.45 (0.43–0.47) | 6890 | 20.2 | 1.00 (ref) |
| 2-<5 years | 2968 | 20.4 | 0.78 (0.75–0.81) | 4183 | 16.7 | 0.68 (0.65–0.70) | 7845 | 23.0 | 1.00 (ref) |
| 5-<10 years | 4731 | 32.5 | 1.11 (1.08–1.15) | 8741 | 34.9 | 1.20 (1.17–1.23) | 9963 | 29.2 | 1.00 (ref) |
| 10+ years | 4405 | 30.2 | 1.59 (1.55–1.64) | 9358 | 37.3 | 1.87 (1.83–1.92) | 8006 | 23.5 | 1.00 (ref) |
| **Median T2D duration** | 6.7 | (3.3–11.2) | | 8.0 | (4.6–12) | | 5.4 | (2.1–9.6) | 1.00 (ref) |
| **ASCVD** | 4085 | 28.0 | 1.13 (1.10–1.16) | 7241 | 28.9 | 1.09 (1.06–1.12) | 10370 | 30.4 | 1.00 (ref) |
| **Myocardial Infarction** | 2989 | 20.5 | 1.17 (1.12–1.21) | 5328 | 21.3 | 1.13 (1.09–1.16) | 7425 | 21.8 | 1.00 (ref) |
| **Cerebrovascular disease** | 1110 | 7.6 | 0.94 (0.88–1.00) | 1971 | 7.9 | 0.90 (0.86–0.95) | 3590 | 10.5 | 1.00 (ref) |
| **Abdominal and peripheral vascular disease** | 1245 | 8.5 | 1.30 (1.22–1.39) | 2173 | 8.7 | 1.23 (1.17–1.30) | 2841 | 8.3 | 1.00 (ref) |
| **Diabetic retinopathy** | 2640 | 18.1 | 1.20 (1.15–1.25) | 5190 | 20.7 | 1.29 (1.25–1.33) | 6848 | 20.1 | 1.00 (ref) |
| **Diabetic nephropathy** | 1097 | 7.5 | 1.00 (0.94–1.07) | 1548 | 6.2 | 0.77 (0.73–0.82) | 3151 | 9.2 | 1.00 (ref) |
| **Diabetic neuropathy** | 1060 | 7.3 | 1.60 (1.49–1.72) | 2058 | 8.2 | 1.70 (1.60–1.81) | 1881 | 5.5 | 1.00 (ref) |
| **Comorbidity level †** | | | | | | | | | |
| 0 | 9024 | 61.9 | 1.00 (0.98–1.01) | 15909 | 63.5 | 1.06 (1.05–1.07) | 19272 | 56.6 | 1.00 (ref) |
| 1 | 2804 | 19.2 | 1.12 (1.07–1.16) | 4802 | 19.2 | 1.09 (1.05–1.12) | 6263 | 18.4 | 1.00 (ref) |
| 2 | 1603 | 11.0 | 1.08 (1.02–1.14) | 2609 | 10.4 | 0.97 (0.92–1.01) | 4318 | 12.7 | 1.00 (ref) |
| >=3 | 1140 | 7.8 | 0.90 (0.85–0.96) | 1750 | 7.0 | 0.74 (0.70–0.78) | 4226 | 12.4 | 1.00 (ref) |
| **Chronic heart failure** | 914 | 6.3 | 1.06 (0.99–1.14) | 1466 | 5.8 | 0.90 (0.85–0.96) | 2768 | 8.1 | 1.00 (ref) |
| **Atrial fibrillation** | 1113 | 7.6 | 1.09 (1.02–1.16) | 1830 | 7.3 | 0.94 (0.89–0.99) | 3746 | 11.0 | 1.00 (ref) |
| **Hypertension** | 6044 | 41.5 | 1.24 (1.21–1.27) | 10313 | 41.1 | 1.19 (1.16–1.21) | 13385 | 39.3 | 1.00 (ref) |
| **COPD** | 1511 | 10.4 | 1.10 (1.03–1.16) | 2242 | 8.9 | 0.93 (0.88–0.98) | 3551 | 10.4 | 1.00 (ref) |
| **Cancer** | 1208 | 8.3 | 1.01 (0.95–1.07) | 2083 | 8.3 | 0.94 (0.89–0.99) | 3964 | 11.6 | 1.00 (ref) |
| **Renal Disease** | 832 | 5.7 | 0.94 (0.87–1.02) | 1305 | 5.2 | 0.84 (0.78–0.89) | 2176 | 6.4 | 1.00 (ref) |
| **Rheumatic disease** | 518 | 3.6 | 1.09 (0.98–1.20) | 728 | 2.9 | 0.88 (0.81–0.97) | 1260 | 3.7 | 1.00 (ref) |
| **Osteoarthritis** | 2835 | 19.5 | 1.26 (1.21–1.31) | 4643 | 18.5 | 1.15 (1.11–1.19) | 6435 | 18.9 | 1.00 (ref) |
| **Osteoporosis/fracture** | 203 | 1.4 | 0.82 (0.70–0.95) | 332 | 1.3 | 0.72 (0.64–0.82) | 916 | 2.7 | 1.00 (ref) |

*(Continued)*

**Table 1.** (Continued)

| | GLP-1RA | | | SGLT2i | | | DPP-4i | | |
|---|---|---|---|---|---|---|---|---|---|
| | N = 14,671 | Percent (%) | aPR (95% CI) versus DPP-4i § | N = 25,070 | Percent (%) | aPR (95% CI) versus DPP-4i § | N = 34,079 | Percent (%) | |
| **History of infections requiring hospitalization** | 5378 | 36.9 | 1.06 (1.03–1.08) | 8299 | 33.1 | 0.94 (0.92–0.97) | 12116 | 35.6 | 1.00 (ref) |
| **Obesity** | 4468 | 30.7 | 1.60 (1.55–1.66) | 6365 | 25.4 | 1.45 (1.40–1.49) | 5653 | 16.6 | 1.00 (ref) |
| **Alcoholism** | 155 | 1.1 | 0.85 (0.70–1.02) | 233 | 0.9 | 0.71 (0.61–0.83) | 444 | 1.3 | 1.00 (ref) |
| **Mental Disorders** | 7973 | 54.7 | 1.05 (1.03–1.07) | 13096 | 52.2 | 1.01 (1.00–1.03) | 17767 | 52.1 | 1.00 (ref) |
| **Previous hypoglycaemia** | 163 | 1.1 | 1.10 (0.92–1.31) | 261 | 1.0 | 0.96 (0.83–1.12) | 479 | 1.4 | 1.00 (ref) |
| **Trombocyte aggregation prophylaxis** | 5091 | 34.9 | 1.18 (1.15–1.21) | 9630 | 38.4 | 1.22 (1.20–1.25) | 12306 | 36.1 | 1.00 (ref) |
| **Statins** | 10672 | 73.2 | 1.07 (1.05–1.08) | 19212 | 76.6 | 1.10 (1.09–1.11) | 24466 | 71.8 | 1.00 (ref) |
| **ACE inhibitors** | 5425 | 37.2 | 1.08 (1.05–1.11) | 9529 | 38.0 | 1.07 (1.05–1.10) | 12576 | 36.9 | 1.00 (ref) |
| **ATII antagonists** | 4907 | 33.7 | 1.22 (1.19–1.25) | 8781 | 35.0 | 1.24 (1.21–1.27) | 10385 | 30.5 | 1.00 (ref) |
| **Any antihypertensive treatment** | 11458 | 78.6 | 1.10 (1.09–1.11) | 20024 | 79.9 | 1.09 (1.08–1.10) | 26629 | 78.1 | 1.00 (ref) |
| **Oral steriods** | 909 | 6.2 | 0.94 (0.88–1.01) | 1376 | 5.5 | 0.80 (0.75–0.85) | 2713 | 8.0 | 1.00 (ref) |
| **Marital status** | | | | | | | | | |
| Divorced | 2519 | 17.3 | 1.08 (1.04–1.13) | 4214 | 16.8 | 1.06 (1.03–1.10) | 5461 | 16.0 | 1.00 (ref) |
| Married | 8019 | 55.0 | 1.01 (0.99–1.03) | 14464 | 57.7 | 1.03 (0.98–1.08) | 19011 | 55.8 | 1.00 (ref) |
| Unknown | 178 | 1.2 | 0.83 (0.70–0.99) | 197 | 0.8 | 1.06 (1.02–1.10) | 422 | 1.2 | 1.00 (ref) |
| Unmarried | 2738 | 18.8 | 1.06 (1.01–1.10) | 4196 | 16.7 | 1.04 (1.03–1.06) | 4604 | 13.5 | 1.00 (ref) |
| Widowed | 1117 | 7.7 | 1.08 (1.01–1.14) | 1999 | 8.0 | 0.57 (0.48–0.67) | 4581 | 13.4 | 1.00 (ref) |
| **Median baseline % HbA$_{1c}$ (IQR) ‡** | 8.4 | (7.5–9.5) | | 8.3 | (7.5–9.4) | | 7.8 | (7.2–8.8) | |
| **Median baseline eGFR ml/min/ 1.73m$^2$ (IQR) ‡** | 87 | (62–100) | | 89 | (73–106) | | 82 | (62–100) | |
| **Median baseline LDL mmol/L (IQR) ‡** | 2.0 | (1.5–2.6) | | 1.9 | (1.6–2.5) | | 2.0 | (1.6–2.7) | |

† Charlson Comorbity level calculated as a total score of 0, 1, 2 or 3 and more.

‡ Numbers based on the North and Central Denmark Regions (~30% of total Danish population) where laboratory data were available.

§ Adjusted for differences in age and sex.

compared with GLP-1RA being an established T2D treatment already at the beginning of our study period. Thus, when comparing new users of the three drug classes, SGLT2i increased from the least prescribed drug class per 100,000 and to the most prescribed drug class in only 3 years. Notably, following the positive CV outcome trial EMPA-REG OUTCOME in 2015 Q3 there was a large increase in the overall prescription of SGLT2i, continuing an increasing SGLT2i trend seen already before 2015 Q3, yet primarily related to distinct increase in empa-gliflozin use after 2015 Q3. Since the national diabetes guidelines were not changed during the study time period, this increase could be driven by the published CV outcome trial results and to some extend the following CV label updates. A similar jump in increase was not observed for GLP-1RA or liraglutide following the positive CV outcome trial LEADER in 2016 Q3, i.e., for GLP-1RA the incidence increase over time was more modest and steady. In a sensitivity analysis, inclusion of liraglutide 3.0 mg daily (obesity treatment label) in our incidence analy-ses slightly raised the GLP-1RA initiation incidence trend. In addition to the difference in time at market, the lower rise in new prescriptions among GLP1-RA may also reflect the difference in drug administration (oral vs injectable).

Despite increased use of the newer drug classes (SGLT2i and GLP1-RA) and the docu-mented 10% annual decline in incidence of T2D in Denmark since 2012 [27], the DPP4i class

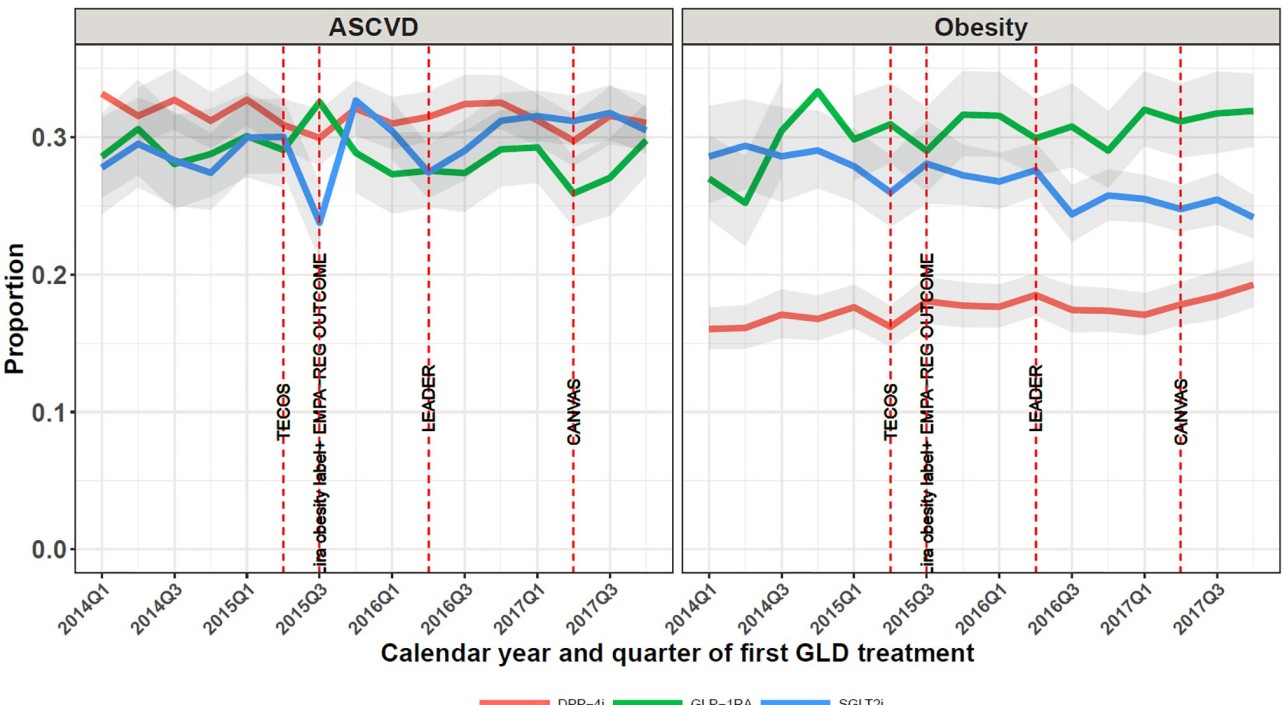

**Fig 3. Time trends in proportions with established atherosclerotic cardiovascular disease (ASCVD) or hospital-diagnosed obesity at baseline.**
Obesity: used hospital inpatient and outpatient contacts. ASCVD: atherosclerotic cardiovascular disease; DPP-4i: dipeptidyl peptidase-4 inhibitor; GLP-1RA: glucagon-like peptide-1 receptor agonists; SGLT2i: sodium-glucose cotransporter 2 inhibitors; TECOS: Sitagliptin (DPP-4i) showed non-inferiority to placebo [35]; Lira obesity label: Liraglutide 3 mg launched as treatment for obesity; EMPA-REG OUTCOME: empagliflozin showed CV and CV/all-cause mortality benefits [10], LEADER: liraglutide showed CV and CV/all-cause mortality benefits [12]; CANVAS: canagliflozin showed CV benefits [11].

initiation did not decline. This may be due to a generally increased focus on timely and earlier treatment intensification with add-on therapies for good glycemic control, supported by updated guidelines [15,28].

As a side finding, we observed that all drug classes saw an incidence decline in all Q3s, presumably an effect from widespread summer vacations in the Danish healthcare system July-August, with expected fewer planned patient therapy changes.

**Treatment lines.** We found that treatment lines of GLP-1RA initators remained relatively stable throughout the period. During this time, the use of SGLT2i as second line therapy increased markedly. This was corroborated by recent US findings by Montvida et al, demonstrating a fast SGLT2i adaptation constituting 7% of all second line drugs used in 2016 versus 0% in 2013, while GLP-1RA as second-line drug increased from 5% to 7% [29].

**CVD prevalence.** In Denmark, 20% to 25% of patients with early type 2 diabetes have ASCVD [30,31]. This is less than observed in either of the drug initiation classes we examined, which may be due to the majority of patients in our study study having a diabetes duration of at least 5 to 8 years together with a clinical decision of adding a subsequent drug class. Following the CV outcome trials for SGLT2i, some of the differences between characteristics of SGLT2i and DPP4i initiators diminished. When adjusting for higher age among DPP-4i initiators (and thus taking into account the expected age-related increase in comorbidities), we did find a slightly higher CV disease prevalence among SGLT2i and GLP-1RA users. Of note however, according to updated treatment guidelines [15,16] also patients with advanced age and CVD would likely benefit from treatment with SGLT2i or GLP-1RA as compared with DPP-

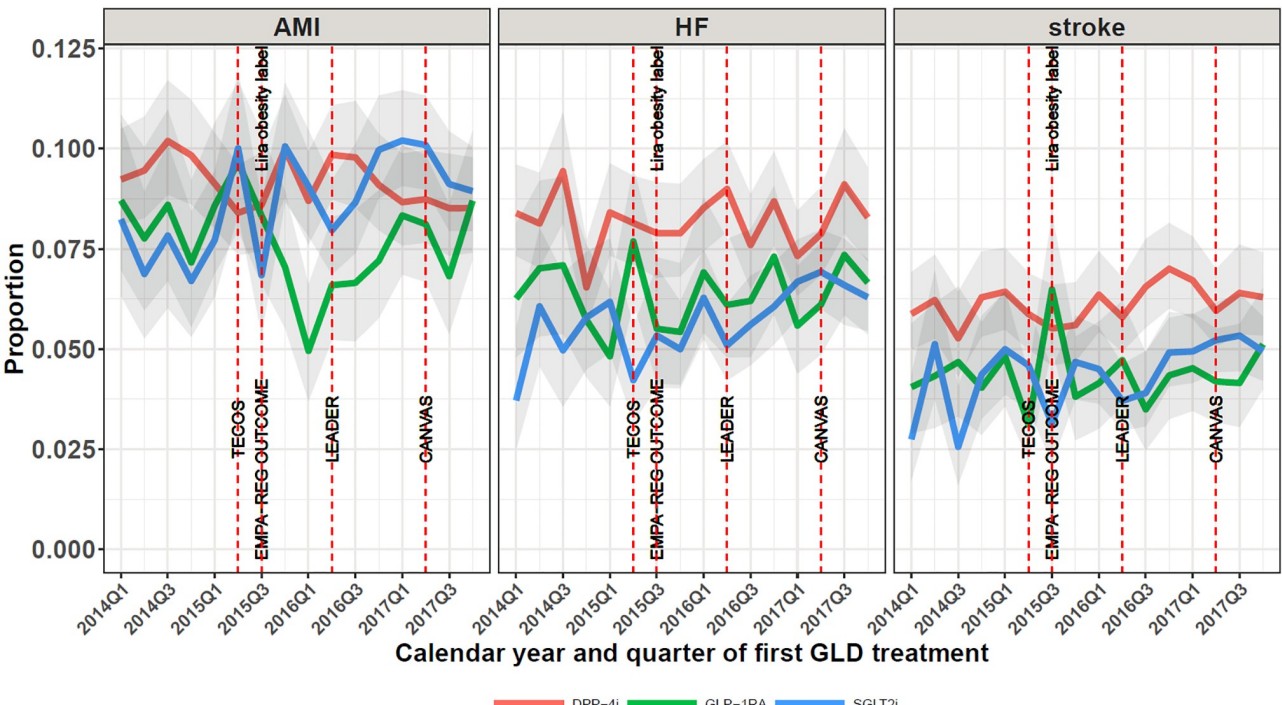

**Fig 4. Time trends in patient proportions with acute myocardial infarction (AMI), heart failure (HF), or stroke at baseline.** AMI: acute myocardial infarction; HF: heart failure; DPP-4i: dipeptidyl peptidase-4 inhibitor; GLP-1RA: glucagon-like peptide-1 receptor agonists; SGLT2i: sodium-glucose cotransporter 2 inhibitors; TECOS: Sitagliptin (DPP-4i) showed non-inferiority to placebo [35]; Lira obesity label: Liraglutide 3 mg launched as treatment for obesity; EMPA-REG OUTCOME: empagliflozin showed CV and CV/all-cause mortality benefits [10], LEADER: liraglutide showed CV and CV/all-cause mortality benefits [12]; CANVAS: canagliflozin showed CV benefits [11].

4i. Our findings are in line with a recent Danish study, finding that presence or absence of previous CVD had little effect on SU prescription likelihood during 2006–2012 despite the potential CVD risk associated with use of SUs [32]. Specifically for GLP-1RA initators, the lower crude prevalence of many CV conditions may in part be attributed to liraglutide being used more often in women related to its weight-reducing effect, with middle-aged women in general having lower risk of CV disease versus men.

## Strengths and limitations

All antihyperglycemic drugs require prescriptions by a physician in Denmark and are partially reimbursed, making our drug utilization coverage close to complete on a population-based nationwide level, minimizing the risk of selection bias in the identification of patients with T2D often seen in other clinic-based studies. We have recently found evidence for high positive predictive values of the CV diagnoses used in the present study (e.g., myocardial infarction: 97%, heart failure: 76%, stroke: 97%) [21,33]. We defined the term ASCVD used in recent diabetes guidelines as either ischemic heart disease (composed of unstable angina [PPV: 46%], myocardial infarction [PPV: 97%] or other ischemic heart disease [unknown PPV]), cerebrovascular disease [PPV: 97%], abdominal or peripheral vascular disease [PPV 100%]), while keeping heart failure [PPV 76%] as a separate category [21,34]. The PPV of obesity diagnosis is unkown but presumably high, although underrecording of obesity is likely. Since any misclassifications of these diseases is unlikely related to which antihyperglycemic drug class a patient uses, this is unlikely to have major impact on our findings.

## Conclusions

The current study provides evidence for some dynamics in the use of SGLT2i and GLP-1RAs (increase in overall use for both drug classes; increase in 2$^{nd}$ line therapy use for SGLT2i), with remarkably little changes by now in the characteristics including CVD prevalences of patients intiating these drugs. If physicians had closely followed the newest clinical trial results and adapted their T2D treatment accordingly, increasing initiation of SGLT2is and GLP-1RAs among T2D patients with prevalent CVD could be expected [15] (with opposite findings for new DPP4i users). Our findings may indicate that physicians caring for T2D patients in real life are impacted by recent new and convenient HbA1c-lowering treatment options, but until now only to a limited extent have considered presence or absence of CVD in their treated patients. Thus, the large increase in SGLT2i and GLP-1RAs utilization may relate to generally increased clinician knowledge and awareness of effectiveness and safety of these newer drug classes in recent years, and thus more confidence in prescribing. Physicians may in addition have considered the drug effect on other clinically relevant parameters such as magnitude of HbA$_{1c}$ reduction, as well as CV risk factors including body weight and blood pressure. It will be interesting to follow how user characteristics may change after the recently updated national [15] and international guidelines [14,16], which clearly guide treatment choices not only based on HbA1c level and CV risk factors including weight and blood pressure, but also more specifically on preexisting CVD including heart failure and kidney impairment.

In conclusion, following the EMPA-REG OUTCOME trial SGLT2i were increasingly used as 2nd-line treatment in everyday clinical practice, with only minor increases in proportions with ASCVD over time. For GLP-1RA, proportions with ASCVD have decreased, despite publication of the LEADER trial. Recently updated guidelines for T2D patients with ASCVD and heart failure/ renal impairment may affect these real-world trends in the future.

## Supporting information

**S1 File.**
(DOCX)

## Author Contributions

**Conceptualization:** Jakob S. Knudsen, Maria Lajer, Larisa Nurkanovic, Anastasia Ustyugova, Henrik T. Sørensen, Reimar W. Thomsen.

**Data curation:** Jakob S. Knudsen, Reimar W. Thomsen.

**Formal analysis:** Jakob S. Knudsen, Lisbeth M. Baggesen, Reimar W. Thomsen.

**Investigation:** Jakob S. Knudsen, Reimar W. Thomsen.

**Methodology:** Jakob S. Knudsen, Lisbeth M. Baggesen, Larisa Nurkanovic, Anastasia Ustyugova, Reimar W. Thomsen.

**Project administration:** Jakob S. Knudsen, Reimar W. Thomsen.

**Supervision:** Henrik T. Sørensen, Reimar W. Thomsen.

**Visualization:** Jakob S. Knudsen.

**Writing – original draft:** Jakob S. Knudsen, Maria Lajer, Anastasia Ustyugova, Reimar W. Thomsen.

**Writing – review & editing:** Jakob S. Knudsen, Lisbeth M. Baggesen, Maria Lajer, Larisa Nurkanovic, Anastasia Ustyugova, Henrik T. Sørensen, Reimar W. Thomsen.

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
