## [Decision Letter · Decision Letter 0]

5 Dec 2019

PONE-D-19-29684

Changes in SGLT2i and GLP-1RA real-world initiator profiles following cardiovascular outcome trials: A Danish nationwide population-based study

PLOS ONE

Dear Dr knudsen,

Thank you for submitting your manuscript to PLOS ONE. After careful consideration, we feel that it has merit but does not fully meet PLOS ONE’s publication criteria as it currently stands. Therefore, we invite you to submit a revised version of the manuscript that addresses the points raised during the review process.

We would appreciate receiving your revised manuscript by Jan 19 2020 11:59PM. To enhance the reproducibility of your results, we recommend that if applicable you deposit your laboratory protocols in protocols.io, where a protocol can be assigned its own identifier (DOI) such that it can be cited independently in the future. For instructions see: http://journals.plos.org/plosone/s/submission-guidelines#loc-laboratory-protocols

We look forward to receiving your revised manuscript.

Kind regards,

Tatsuo Shimosawa, M.D., Ph.D.

Academic Editor

PLOS ONE

Journal Requirements:

2. In ethics statement in the manuscript and in the online submission form, please provide additional information about the patient records used in your retrospective study. Specifically, please ensure that you have discussed whether all data were fully anonymized before you accessed them and/or whether the IRB or ethics committee waived the requirement for informed consent. If patients provided informed written consent to have data from their medical records used in research, please include this information.

3. Thank you for including your competing interests statement; "ML, LN and AU are employees of Boehringer Ingelheim. The other authors have no personal conflicts of interest relevant to this article. The Department of Clinical Epidemiology is involved in studies with funding from various companies as research grants to (and administered by) Aarhus University."

We note that one or more of the authors are employed by a commercial company: name of commercial company.

4. Please ensure that you refer to Figure 4 in your text as, if accepted, production will need this reference to link the reader to the figure.

Reviewers' comments:

Reviewer's Responses to Questions

**Comments to the Author**

1. Is the manuscript technically sound, and do the data support the conclusions?

Reviewer #1: Yes

Reviewer #2: Yes

2. Has the statistical analysis been performed appropriately and rigorously? 

Reviewer #1: Yes

Reviewer #2: Yes

3. Have the authors made all data underlying the findings in their manuscript fully available?

Reviewer #1: Yes

Reviewer #2: No

4. Is the manuscript presented in an intelligible fashion and written in standard English?

Reviewer #1: No

Reviewer #2: Yes

5. Review Comments to the Author

Reviewer #1: This paper, describing prescription patterns of anti diabetic drugs, presents new research of both local and general interest. It is methodologically sound and falls within the scope of PLOSone.

In brief, the authors focus on the impact of recent CVOTs demonstrating cardioprotective effects of SGLT-2 inhibitors and GLP-1 analogues. They describe first use of these drugs according to patient characteristics (a stable prevalence of CVD)and the appearance of publications and conclude that, in particular the EMPAreg study, CVOTs impact prescription patterns.

A few points for the author's considerations:

Is the impact of EMPA-reg really substantiated for the use of SGLT-2 inhibitors as stated (the slope of the curve in Fig1 appears to be exactly the same before and after publication of the EMPAreg)? The shift seems to be a larger proportion of empagliflozin (vs dapagliflozin).

The authors might briefly discuss the original design and intended impact of CVOTs (primarily safety trials) and maybe also state that a negative impact of trials can be seen as well; the authors themselves have described negative impacts of trials on prescription patterns (BMJ Open. 2013 Sep 24;3(9):e003424).

The total number of users of GLP-1 analogues and SGLT2 inhibitors during the period would be of interest.

Since previous CVD does not appear to be the driver for prescription of drugs with a potential benefit (well-known also for SUs, Basic Clin Pharmacol Toxicol. 2018 Jun;122(6):606-61), could the authors identify or even quantify more marketing-related mechanisms which would most likely also influence the pattern?

Finally, check the language, e.g. a lot of of's in lines 17-18, p3.

Reviewer #2: Jakob et al. present nationwide changes in clinical characteristics of initiators of SGLT2i and GLP-1RA, the relatively new class of medicine for type 2 diabetes treatment, and found an increased usage of SGLT2i, especially empagliflozin, around the publish of EMPAREG-OUTCOME trial, while GLP-1RA therapy lines are stable around LEADER trial. Moreover, the authors found that improved CV outcomes in those trials did NOT lead to increase the proportion of target patient from the trial. The study is well-designed, including the enough number of patients, and of interest for the researchers in the field of diabetics and economics. However, there are several concerns to clarify before published.

Major

1. In p12 author discuss, “Notably, following the positive CV outcome trial EMPA-REG OUTCOME in 2015 Q3 there was a distinct and large increase in the prescription of SGLT2is”

However, the tendency of the whole SGLT2is’ prescriptions between before and after the publish of EMPAREG-OUTCOME trial seems not so changed in the figure. SGLT2is’ initiators seem linearly increasing throughout 2014-2017. As authors pointed out in p13, an incidence decline in all Q3s makes difficult to read the effect of EMPAREG-OUTCOME. So, proof by quantitative evaluation between before and after initiation the trial is needed. Causal relationship might be difficult to be proved, but that data helps our interpretation. e.g. if the number of SGLT2is initiation is not relating to that trial, there are no wonder even the proportion of ASCVD is not increased.

2. Is there any change in ASCVD proportion between empagliflozin and other SGLT2is initiators after the EMPAREG-OUTCOME trial?

3. In Figure3, there is a drop down about ASCVD proportion of SGLT2i initiators and reciprocal increase of GLP-1RA in 2015Q3. Isn’t there any comment?

4. In Japan (my country), there is an alert from the committee of japan diabetes society for the risk of SGLT2i, e.g. about dehydration leading to cerebrovascular infarction, especially in the elderly. I think the result of EMPAREG-OUTCOME gave the great relief on the doctors to prescribe widely (although it is mere speculation) because even patients with the most CV risks are safe for the SGLT2i initiation. Please describe the policies and guidelines applied to Danish diabetic patients concerning the study, for readers outside Denmark.

Minor

Please check the following

p1 Author name jakob s ->Jakob S.

p11 (95%CI 1.01.1.19)->1.01-1.19

6. PLOS authors have the option to publish the peer review history of their article (what does this mean?). If published, this will include your full peer review and any attached files.

Reviewer #1: No

Reviewer #2: No

---

## [Author Response · Author response to Decision Letter 0]

17 Jan 2020

RESPONSE LETTER

Response: 

We have adjusted the manuscript in accordance with PLOS ONE style guidelines. 

2. In ethics statement in the manuscript and in the online submission form, please provide additional information about the patient records used in your retrospective study. Specifically, please ensure that you have discussed whether all data were fully anonymized before you accessed them and/or whether the IRB or ethics committee waived the requirement for informed consent. If patients provided informed written consent to have data from their medical records used in research, please include this information.

Response: 

The ethics statement now reads:

“The study was approved by the Danish Data Protection Agency. Analyses were conducted on pseudonymized data at the Danish Health Data Authority. The study was purely registry-based and did not involve any contact with patients or interventions; therefore, according to Danish legislation, no informed consent or approval from the health research ethics committee was required.”

3. Thank you for including your competing interests statement; "ML, LN and AU are employees of Boehringer Ingelheim. The other authors have no personal conflicts of interest relevant to this article. The Department of Clinical Epidemiology is involved in studies with funding from various companies as research grants to (and administered by) Aarhus University."

We note that one or more of the authors are employed by a commercial company: name of commercial company. 

Response: 

The statement now reads:

“Sources of funding: This work was supported by an institutional research grant from Boehringer Ingelheim to Aarhus University. While Boehringer Ingelheim was involved in the study’s concept, it was designed and conducted by the coauthors from Aarhus University. The funder provided support in the form of salaries for authors ML, LN, and AU, but did not have any additional role in the study design, data collection and analysis, decision to publish, or preparation of the manuscript. The specific roles of these authors are articulated in the ‘author contributions’ section. “

Response: 

Please see our response above.

Response:

Please see our response above.

 Response: 

We have included an amended text reading: 

“Conflict of interest disclosures: ML, LN and AU are employees of Boehringer Ingelheim. The other authors have no personal conflicts of interest relevant to this article. The Department of Clinical Epidemiology is involved in studies with funding from various companies as research grants to (and administered by) Aarhus University. The commercial affiliation did not alter our adherence to PLOS ONE policies on sharing data and materials. However, Danish law does not allow researchers to share raw data from the registries with third parties. Data can be accessed by researchers through application to the Danish Data Protection Agency and the Danish Health Data Authority.”

Response: 

Please see our response above.

4. Please ensure that you refer to Figure 4 in your text as, if accepted, production will need this reference to link the reader to the figure.

Response:

The manuscript reads: “Fig 4 shows time trends from 2014 Q1 to 2017 Q4 in the proportion of SGLT2i, GLP-1RA, and DPP-4i initiators who had acute myocardial infarction (AMI) and heart failure (HF), respectively, diagnosed prior to treatment initiation.”

Response: 

We have included captions for Supporting Information at the end of the manuscript. We have renamed and renumbered tables and figures in the supplementary and updated the manuscript accordingly.

Review Comments to the Author

Reviewer #1: This paper, describing prescription patterns of anti diabetic drugs, presents new research of both local and general interest. It is methodologically sound and falls within the scope of PLOSone.

Response: 

We thank the reviewer for finding our study both interesting and sound.

In brief, the authors focus on the impact of recent CVOTs demonstrating cardioprotective effects of SGLT-2 inhibitors and GLP-1 analogues. They describe first use of these drugs according to patient characteristics (a stable prevalence of CVD) and the appearance of publications and conclude that, in particular the EMPAreg study, CVOTs impact prescription patterns.

A few points for the author's considerations:

Is the impact of EMPA-reg really substantiated for the use of SGLT-2 inhibitors as stated (the slope of the curve in Fig1 appears to be exactly the same before and after publication of the EMPAreg)? The shift seems to be a larger proportion of empagliflozin (vs dapagliflozin).

Response: 

We fully agree with the reviewer that, when considering fig 1 in isolation, it is visually less clear if there is a major change in the prescription incidence trend of SGLT2i overall. When considering S2 fig, we believe it more clearly shows an increase in the slope in SGLT2i prescription trends, solely driven by large increases in empagliflozin initiation. 

We have now modified the statements in the manuscript describing the changes following EMPA-REG outcome. As we wrote on page 14, “Notably, following the positive CV outcome trial EMPA-REG OUTCOME in 2015 Q3 there was a distinct and large increase in the prescription of SGLT2is, primarily related to increase in empagliflozin use.”.

The authors might briefly discuss the original design and intended impact of CVOTs (primarily safety trials) and maybe also state that a negative impact of trials can be seen as well; the authors themselves have described negative impacts of trials on prescription patterns (BMJ Open. 2013 Sep 24;3(9):e003424).

Response:

We thank the reviewer for having read the previous publication and for pointing out that the purpose of CVOTs may not be fully clear from the manuscript. We have updated the manuscript accordingly:

“In the most recent years, the prescription patterns of SGLT2i and GLP-1RAs in real-world settings may have been increasingly influenced by landmark cardiovascular (CV) outcome trials instigated by regulatory authorities to promote patient safety [9],”

The total number of users of GLP-1 analogues and SGLT2 inhibitors during the period would be of interest.

Response: 

We agree. We have now included these data in the beginning of the Results:

“We identified 25,070 new first-time SGLT2i initiators, 14,671 first-time GLP-1RA initiators, and 34,079 first-time DPP-4i initiators in Denmark during 2014-2017. As a point of comparison, the total number of unique prevalent users in Denmark during 2014-2017 was 32,091 for SGLT2i, 37,282 for GLP-1RA, and 64,613 for DPP-4i (data not shown).”

Since previous CVD does not appear to be the driver for prescription of drugs with a potential benefit (well-known also for SUs, Basic Clin Pharmacol Toxicol. 2018 Jun;122(6):606-61), could the authors identify or even quantify more marketing-related mechanisms which would most likely also influence the pattern?

Response:

We thank the reviewer for pointing out the interesting study in BCPT, which may corroborate our findings that clinical considerations about pre-existing CVD or not may not be the main or only driver for the choice of prescription.

We have updated our discussion section with the reference and a remark including this important aspect. “Our findings are in line with a recent Danish study, finding that presence or absence of previous CVD had little effect on SU prescription likelihood during 2006-2012 despite the potential CVD risk associated with use of SUs [32].”

Identifying the exact marketing-related mechanisms that are most likely to have resulted in the increase in adaptation would indeed be valuable, however, it is beyond the scope of this paper, and unfortunately also beyond the data available in our dataset.

We have included the following sentence: “Thus, the large increase in SGLT2i and GLP-1RAs utilization may relate to generally increased clinician knowledge and awareness of effectiveness and safety of these newer drug classes in recent years, and thus more confidence in prescribing.”

Finally, check the language, e.g. a lot of of's in lines 17-18, p3.

Response:

We thank the reviewer for pointing this out and have revised the language.

Reviewer #2: Jakob et al. present nationwide changes in clinical characteristics of initiators of SGLT2i and GLP-1RA, the relatively new class of medicine for type 2 diabetes treatment, and found an increased usage of SGLT2i, especially empagliflozin, around the publish of EMPAREG-OUTCOME trial, while GLP-1RA therapy lines are stable around LEADER trial. Moreover, the authors found that improved CV outcomes in those trials did NOT lead to increase the proportion of target patient from the trial. The study is well-designed, including the enough number of patients, and of interest for the researchers in the field of diabetics and economics. 

Response: 

We thank the reviewer for finding our study well-designed and interesting.

However, there are several concerns to clarify before published.

Major

1. In p12 author discuss, “Notably, following the positive CV outcome trial EMPA-REG OUTCOME in 2015 Q3 there was a distinct and large increase in the prescription of SGLT2is”

However, the tendency of the whole SGLT2is’ prescriptions between before and after the publish of EMPAREG-OUTCOME trial seems not so changed in the figure. SGLT2is’ initiators seem linearly increasing throughout 2014-2017. As authors pointed out in p13, an incidence decline in all Q3s makes difficult to read the effect of EMPAREG-OUTCOME. So, proof by quantitative evaluation between before and after initiation the trial is needed. Causal relationship might be difficult to be proved, but that data helps our interpretation. e.g. if the number of SGLT2is initiation is not relating to that trial, there are no wonder even the proportion of ASCVD is not increased.

Response:

We fully agree with the reviewer that, when considering fig 1 in isolation, it is visually less clear if there is a major change in the prescription incidence trend of SGLT2i overall after 2015, or if there is merely a continuously increase in SGLT2i incidence throughout the entire study period.

When considering S2 fig, we believe it more clearly shows an increase in the slope in SGLT2i prescription trends, solely driven by large increases in empagliflozin initiation after 2015.

We have now modified the statements in the manuscript describing the changes following EMPA-REG outcome, to make them more precise. “Notably, following the positive CV outcome trial EMPA-REG OUTCOME in 2015 Q3 there was a large increase in the overall prescription of SGLT2i, continuing an increasing SGLT2i trend seen already before 2015 Q3, yet primarily related to distinct increase in empagliflozin use after 2015 Q3.”

We also agree that SGLT2i trends obviously may be affected by other factors than a CVOT, and we have now included the following note in the Discussion section: “Thus, the large increase in SGLT2i and GLP-1RAs utilization may relate to generally increased clinician knowledge and awareness of effectiveness and safety of these newer drug classes in recent years, and thus more confidence in prescribing.”

2. Is there any change in ASCVD proportion between empagliflozin and other SGLT2is initiators after the EMPAREG-OUTCOME trial?

Response:

Below we have created the figure 3 ASCVD panel, restricted to SGLT2i and stratified by individual drug types in order to address this question. We observe a slight increase since the time of the EMPAREG OUTCOME trial in the proportion with ASCVD for both dapagliflozin and empagliflozin initiators (canagliflozin starters are too few to yield statistically reliable trends). The ASCVD proportion was about 5 percentage points higher in empagliflozin than in dapagliflozin starters. This ASCVD difference between the 2 drugs was rather constant over time, and the difference was visible already before the EMPAREG outcome trial and not clearly affected by the trial publication.

We have added to the manuscript Results: “A slight increase in ASCVD since 2015 was seen in parallel in the two major groups of empagliflozin and dapagliflozin initiators, with the ASCVD proportion continuously being about 5 percentage points higher in empagliflozin than in dapagliflozin starters (data not shown).

3. In Figure3, there is a drop down about ASCVD proportion of SGLT2i initiators and reciprocal increase of GLP-1RA in 2015Q3. Isn’t there any comment?

Response:

We agree with the reviewer that these apparent “outliers” in 2015Q3 seem a bit remarkable, especially considering the timing co-incident with the EMPA-REG outcome trial publication. We have also discussed this finding. Both curves (for GLP-1RA and SGLT2i) in the following quarter show compensatory ASCVD changes back to more “normal” values, leading to ASCVD proportions that – if taking an average over the six months (2015-Q3+2015-Q4) - would be completely in alignment with the overall curve trends. Moreover, we find it less clinically plausible that, over just half a year, we would see first temporary pre-trial changes that are then fully reversed in the following quarter, all as a consequence of the trial. There is some statistical variation seen from quarter to quarter, and we think the “extreme” values coinciding with one data point in 2015Q3 should probably be regarded as outliers, with some regression towards the mean observed in the following quarters.

4. In Japan (my country), there is an alert from the committee of japan diabetes society for the risk of SGLT2i, e.g. about dehydration leading to cerebrovascular infarction, especially in the elderly. I think the result of EMPAREG-OUTCOME gave the great relief on the doctors to prescribe widely (although it is mere speculation) because even patients with the most CV risks are safe for the SGLT2i initiation. Please describe the policies and guidelines applied to Danish diabetic patients concerning the study, for readers outside Denmark.

Response:

We thank the reviewer for pointing out that it is not unambiguously clear from the manuscript that the Danish guidelines has been and are following the international guidelines closely. We have updated the manuscript to reflect this. The manuscript now reads:

“While metformin has remained the recommended initial antihyperglycemic drug for most patients with T2D, international (and Danish) guidelines until now have recommended a free choice among several second or third line treatment options, based on an individualised treatment approach [8]. In the most recent years, the prescription patterns of SGLT2i and GLP-1RAs in real-world settings may have been increasingly influenced by landmark cardiovascular (CV) outcome trials instigated by regulatory authorities to promote patient safety [9], but data on incident utilization trends are scarce. In 2015, the empagliflozin EMPA-REG OUTCOME trial [10] showed a reduced risk of CV outcomes, CV mortality, and all-cause mortality in patients with T2D with existing CV disease. In 2017, the canagliflozin CANVAS trial program [11] showed a reduced risk of major adverse CV events in patients with T2D and high CV risk. For GLP-1RAs, the 2016 liraglutide LEADER trial showed a reduced risk of CV outcomes, CV mortality, and all-cause mortality in patients with high CV risk. A reduced risk of CV outcomes was also observed in similar patients receiving semaglutide in the SUSTAIN-6 trial in 2016 [12,13]. Therefore, in the most recent updates to the EASD/ADA and Danish guidelines from 2018 and 2019 [14–16], initiation of a SGLT2i or a GLP-1RA with proven CV benefit has been recommended for patients with T2D and clinical CV disease, with currently strongest evidence available for liraglutide and empagliflozin [14–16].”

Minor

Please check the following

p1 Author name jakob s ->Jakob S.

p11 (95%CI 1.01.1.19)->1.01-1.19

Response: We thank the reviewer and have corrected the errors.

---

## [Decision Letter · Decision Letter 1]

30 Jan 2020

PONE-D-19-29684R1

Changes in SGLT2i and GLP-1RA real-world initiator profiles following cardiovascular outcome trials: A Danish nationwide population-based study

PLOS ONE

Dear Dr knudsen,

Thank you for submitting your manuscript to PLOS ONE. After careful consideration, we feel that it has merit but does not fully meet PLOS ONE’s publication criteria as it currently stands. Therefore, we invite you to submit a revised version of the manuscript that addresses the points raised during the review process.

We would appreciate receiving your revised manuscript by Mar 15 2020 11:59PM. To enhance the reproducibility of your results, we recommend that if applicable you deposit your laboratory protocols in protocols.io, where a protocol can be assigned its own identifier (DOI) such that it can be cited independently in the future. For instructions see: http://journals.plos.org/plosone/s/submission-guidelines#loc-laboratory-protocols

We look forward to receiving your revised manuscript.

Kind regards,

Tatsuo Shimosawa, M.D., Ph.D.

Academic Editor

PLOS ONE

Journal Requirements:

Additional Editor Comments (if provided):

It is our policy to show all the data, data not shown is not acceptable.

Reviewers' comments:

Reviewer's Responses to Questions

**Comments to the Author**

1. If the authors have adequately addressed your comments raised in a previous round of review and you feel that this manuscript is now acceptable for publication, you may indicate that here to bypass the “Comments to the Author” section, enter your conflict of interest statement in the “Confidential to Editor” section, and submit your "Accept" recommendation.

Reviewer #1: All comments have been addressed

Reviewer #2: All comments have been addressed

2. Is the manuscript technically sound, and do the data support the conclusions?

Reviewer #1: Yes

Reviewer #2: Yes

3. Has the statistical analysis been performed appropriately and rigorously? 

Reviewer #1: Yes

Reviewer #2: Yes

4. Have the authors made all data underlying the findings in their manuscript fully available?

Reviewer #1: Yes

Reviewer #2: No

5. Is the manuscript presented in an intelligible fashion and written in standard English?

Reviewer #1: Yes

Reviewer #2: Yes

6. Review Comments to the Author

Reviewer #1: (No Response)

Reviewer #2: The authors improved the article and adequately discussed all the query.

The only issue left is about data, because "data not shown" is not permitted in PLOS One, shown in https://journals.plos.org/plosone/s/data-availability.

Therefore the authors should revise them.

7. PLOS authors have the option to publish the peer review history of their article (what does this mean?). If published, this will include your full peer review and any attached files.

Reviewer #1: No

Reviewer #2: Yes: Akifumi Kushiyama

---

## [Author Response · Author response to Decision Letter 1]

6 Feb 2020

Please find attached the revised manuscript. The term “Data not shown” was previously used in two places in the manuscript. In order to comply with the guidelines we have revised the manuscript.

The first instance we have simply revised the manuscript that we have made a comparison using an analysis on our own data. The manuscript now reads:

“As a point of comparison, the total number of unique prevalent users in our data during 2014-2017 was 32,091 for SGLT2i, 37,282 for GLP-1RA, and 64,613 for DPP-4i.”

In the second instance, we referred to an analysis performed and presented during the review process, but not included in the manuscript previously. We have now revised the manuscript, adding this to the supplementary and updating the remaining documents accordingly.

---

## [Decision Letter · Decision Letter 2]

11 Feb 2020

Changes in SGLT2i and GLP-1RA real-world initiator profiles following cardiovascular outcome trials: A Danish nationwide population-based study

PONE-D-19-29684R2

Dear Dr. knudsen,

We are pleased to inform you that your manuscript has been judged scientifically suitable for publication and will be formally accepted for publication once it complies with all outstanding technical requirements.

With kind regards,

Tatsuo Shimosawa, M.D., Ph.D.

Academic Editor

PLOS ONE

Additional Editor Comments (optional):

Reviewers' comments:

Reviewer's Responses to Questions

**Comments to the Author**

1. If the authors have adequately addressed your comments raised in a previous round of review and you feel that this manuscript is now acceptable for publication, you may indicate that here to bypass the “Comments to the Author” section, enter your conflict of interest statement in the “Confidential to Editor” section, and submit your "Accept" recommendation.

Reviewer #2: All comments have been addressed

2. Is the manuscript technically sound, and do the data support the conclusions?

Reviewer #2: Yes

3. Has the statistical analysis been performed appropriately and rigorously? 

Reviewer #2: Yes

4. Have the authors made all data underlying the findings in their manuscript fully available?

Reviewer #2: Yes

5. Is the manuscript presented in an intelligible fashion and written in standard English?

Reviewer #2: Yes

6. Review Comments to the Author

Reviewer #2: (No Response)

7. PLOS authors have the option to publish the peer review history of their article (what does this mean?). If published, this will include your full peer review and any attached files.

Reviewer #2: Yes: Akifumi Kushiyama

---

## [Editor Report · Acceptance letter]

18 Feb 2020

PONE-D-19-29684R2 

Changes in SGLT2i and GLP-1RA real-world initiator profiles following cardiovascular outcome trials: A Danish nationwide population-based study 

Dear Dr. knudsen:

I am pleased to inform you that your manuscript has been deemed suitable for publication in PLOS ONE. Congratulations! Your manuscript is now with our production department. 

With kind regards,

on behalf of

Prof. Tatsuo Shimosawa 

Academic Editor

PLOS ONE